

# MAX-DOAS NO₂ observations over Guangzhou, China; ground-based and satellite comparisons

Theano Drosoglou[1], Maria Elissavet Koukouli[1], Natalia Kouremeti[2], Alkiviadis F. Bais[1], Irene Zyrichidou[1], Dimitris Balis[1], Ronald J. van der A[3,4], Jin Xu[5], Ang Li[5]

[1]Laboratory of Atmospheric Physics, Aristotle University of Thessaloniki, Thessaloniki, Greece
[2]Physikalisch-Meteorologisches Observatorium Davos, World Radiation Center, Davos, Switzerland
[3]Royal Netherlands Meteorological Institute (KNMI), De Bilt, the Netherlands
[4]Nanjing University of Information Science and Technology, Nanjing, P.R. China
[5]Anhui Institute of Optics and Fine Mechanics Academy of Sciences (AIOFM), Chinese Academy of Sciences (CAS), Hefei, China

*Correspondence to*: Theano Drosoglou (tdroso@auth.gr)

**Abstract.** In this study, the tropospheric NO₂ vertical column density (VCD) over an urban site in Guangzhou megacity in China is investigated, by means of MAX-DOAS measurements during a campaign from late March 2015 to mid-March 2016. A MAX-DOAS system was deployed at the Guangzhou Institute of Geochemistry of the Chinese Academy of Sciences and operated there for about one year, during the spring and summer months. The tropospheric NO₂ VCDs retrieved by the MAX-DOAS are presented and compared with space-borne observations from GOME-2/MetOp-A, GOME-2/MetOp-B and OMI/Aura satellite sensors. The comparisons reveal good agreement between satellite and MAX-DOAS observations over Guangzhou, with correlation coefficients ranging between 0.76 for GOME-2B and 0.99 for GOME-2A. However, the tropospheric NO₂ loadings are underestimated by the satellite sensors on average by 25.1%, 10.3% and 5.7%, respectively for OMI, GOME-2A and GOME-2B. Our results indicate that GOME-2B retrievals are closer to those of the MAX-DOAS instrument due to the lower tropospheric NO₂ concentrations during the days with valid GOME-2B observations. In addition, the effect of the main coincidence criteria is investigated, namely the cloud fraction (CF), the distance (d) between the satellite pixel center and the ground-based measurement site, as well as the time period within which the MAX-DOAS data are averaged around the satellite overpass time. The effect of CF and time window criteria is more profound on the selection of OMI overpass data, probably due to its smaller pixel size. The available data pairs are reduced to half and about one third for CF≤0.3 and CF≤0.2, respectively, while, compared to larger CF thresholds, the correlation coefficient is improved to 0.99 from about 0.6, the slope value is almost doubled (~0.8) and the mean satellite underestimation is reduced to about half (from ~7 to ~3.5×10¹⁵ molecules/cm²). On the other hand, the distance criterion affects mostly GOME-2B data selection, because GOME-2B pixels are quite evenly distributed among the different radii used in the sensitivity test. More specifically, the number of collocations is notably reduced when stricter radius limits are applied, the r value is improved from 0.76 (d≤50 km) to 0.93 (d≤20 km), and the absolute mean bias decreases about 6 times for d≤30 km compared to the reference case (d≤50 km).



# 1 Introduction

Nitrogen dioxide ($NO_2$) is an important trace gas in the atmosphere. It plays a critical role in the tropospheric photochemistry (Seinfeld and Pandis, 1998; Finlayson-Pitts and Pitts, 2000) and contributes to the radiative forcing of the atmosphere (Solomon et al., 1999). Additionally, $NO_2$ affects the human health causing respiratory problems and is one of the main air pollutants with National Ambient Air Quality Standard (NAAQS) (EPA, 2010; WHO, 2013). It is emitted both by natural and anthropogenic sources; in the first category lie lightning (Schumann and Huntrieser, 2007), agricultural fertilization and the use of nitrogen fixing plants (Vinken et al., 2014 and references therein) and biomass burning (Mebust et al., 2014). In the latter category lie fossil fuel and biofuel combustion, power plant and industrial emissions, ground and air transport, and so on (Olivier and Berdowski, 2001).

The rapid growth of the Chinese economy during the last decades has led to an increase in emissions of air pollutants. Air quality in Chinese megacities has been of great concern in the atmospheric and environmental science community. $NO_2$ is an important trace gas in the troposphere in Chinese megacities (Richter et al., 2005; Ma et al., 2013; Jin et al., 2016) and there is significant evidence that secondary aerosols formed from NOx, as well as $SO_2$ and VOCs, contribute to haze pollution events which are frequently observed over urban agglomerations in China (Fu et al., 2014; Jiang et al., 2015). The investigation of the global and regional spatial gradients and temporal variations of trace gases and the identification of their main emission sources can lead to a better understanding of the haze pollution events and the mechanisms forcing them, offering a useful tool for governments and policy makers in planning and implementing control regulations (Liu et al., 2013).

Guangzhou is the capital of the province of Guangdong in south-eastern China. It is the third most populous city in China, with Shanghai and Beijing being the first two, and one of the most populated metropolitan agglomerations globally. It is located on the Pearl River Delta (PRD) about 120 km north-northwest of Hong Kong. PRD is one of the most economically developed regions in China and one of the largest urban areas and it includes nine cities with a combined population of about 60 million. It is a heavily industrialized area and a major port serving as a transportation and trade hub. PRD suffers from poor air quality and visibility due to rapid industrialization, massive increase in vehicle population and, also, transportation of air pollutants from the nearby cities of Hong Kong and Macau (Wang et al., 2005; Guo et al., 2009). Air quality in the PRD region is characterized by high concentration levels of primary pollutants, such as NOx and $SO_2$, as well as by secondary air pollutants, e.g. ozone and fine particulate matter (Chan and Yao, 2008; Wang et al., 2008; Huang et al., 2012). Shao et al., 2009 amply demonstrated the significant contribution of high NOx levels to the formation of ground-level ozone.

Due to its important role as an air quality indicator, $NO_2$ has been observed and monitored from space-borne instruments for the past three decades. Although a rapid growth in NOx emissions has been observed over China by satellite sensors during the previous two decades (Zhang et al., 2007; Liu et al., 2017), a sharp decline is evident in recent years (Liu et al., 2017; van der A et al., 2017). Satellite observations constitute an important tool of investigating the air pollution levels and trends in global (e.g. Velders et al., 2001; Schneider et al., 2012) and regional (e.g. Zyrichidou et al, 2009; Hilbol et al., 2013) scales. However, the satellite data retrieval is subject to several uncertainty sources related to the spectra analysis and the air mass



factor (AMF) calculation which affect the retrievals of the low tropospheric atmospheric content. The errors introduced by the AMF calculation can be attributed to the a priori profile, the aerosol and cloud properties and the surface albedo assumed (Boersma et al., 2004, 2011; Leitão et al., 2010; Heckel et al., 2011; Lin et al., 2014, 2015). Several validation studies show significant underestimation of tropospheric trace gases, such as $NO_2$, from satellite observations over regions with strong

spatial gradients in tropospheric pollution (e.g. Celarier et al., 2008; Kramer et al., 2008; Chen et al., 2009; Irie et al., 2012; Ma et al, 2013). Considering that the $NO_2$ is distributed mainly in the planetary boundary layer (PBL), well-established ground-based measurements of tropospheric vertical columns and profiles of $NO_2$ are essential for the validation and, subsequently, the improvement of satellite retrievals.

The Multi-Axis Differential Optical Absorption Spectroscopy (MAX-DOAS) (Platt, 1994; Van Roozendael et al., 2003;
Hönninger et al., 2004; Wagner et al., 2004; Wittrock et al., 2004; Platt and Stutz, 2008) is a widely used ground-based remote sensing technique for the retrieval of the vertical column and distribution of various trace gases as well as aerosol properties with relatively high sensitivity in the lower atmosphere (Frieß et al., 2006, 2011, 2016; Clémer et al., 2010; Irie et al., 2008, 2011; Wagner et al., 2011). Moreover, MAX-DOAS measurements have been extensively used for the validation of satellite products (e.g. Brinksma et al., 2008; Herman et al., 2009; Li et al., 2013; Hendrick et al., 2014; De Smedt et al., 2015; Theys
et al., 2015; Jin et al., 2016).

Several studies have validated satellite $NO_2$ products over North China and the Yangtze River delta region using ground-based observations (e.g. Ma et al., 2013; Chan et al., 2015; Jin et al., 2016; Wang et al., 2017b) or have used the satellite measurements of $NO_2$ to estimate NOx emissions (e.g. Ding et al., 2015; Han et al., 2015). However, there are only a few studies for the Pearl River Delta area (e.g. Wu et al., 2013). Within the framework of the EU FP7 Monitoring and Assessment
of Regional air quality in China using space Observations, Project Of Longterm sino-european co-Operation, MarcoPolo project, a MAX-DOAS system was installed by Aristotle University of Thessaloniki (AUTH) in Guangzhou and operating there for about one year. In this study, the tropospheric $NO_2$ vertical column densities derived by the MAX-DOAS are presented and compared with tropospheric $NO_2$ retrievals from OMI/Aura, GOME-2/MetOp-A and GOME-2/MetOp-B satellites.

## 2 MAX-DOAS observations and satellite data sets

### 2.1 Instrumentation and data analysis

A mini MAX-DOAS system (Phaethon) was deployed on the roof of a nine-storey building of the Guangzhou Institute of Geochemistry of the Chinese Academy of Sciences (GIGCAS), China (23°8'54"N; 113°21'32"E; Fig. 1) and was operating there from late March 2015 to mid-March 2016. The instrument comprises a thermoelectrically cooled miniature CCD
spectrograph which detects the radiation in the wavelength range ~300-450 nm with a resolution of about 0.35 nm and acquires fast spectral measurements of both direct solar light and sky radiance. The prototype system was developed in 2006 at the Laboratory of Atmospheric Physics of the Aristotle University of Thessaloniki (LAP-AUTH), Greece (Kouremeti et al., 2008,



2013). Currently, there are three MAX-DOAS systems routinely operating in the greater area of Thessaloniki, Greece. Their operation and their capability in retrieving the tropospheric $NO_2$ have been tested successfully under different air pollution conditions and $NO_2$ loadings (Drosoglou et al., 2017).

Guangzhou is the largest city located in Pearl River Delta region and it is affected from elevated concentrations of NOx (e.g.
Zhou et al., 2007; Chan and Yao, 2008). Guangzhou is characterized by humid subtropical monsoon climate and suffers from occasional typhoons and frequent afternoon thunderstorms during the period from early March to mid-October. Under such weather conditions the instrument operation should be interrupted and the outdoors part of the system should be dismounted and brought indoors. This resulted in significant gaps in the data series of $NO_2$. In addition, the instrument was not operating from late August 2015 to late February 2016, due to an accidental damage of the optical fiber, and subsequently due to problems
in the remote access of the system which was essential for controlling the operation of the instrument. Nevertheless, the MAX-DOAS observations of tropospheric $NO_2$ were quite sufficient to be used for the satellite data validation.

In Guangzhou the system was performing sky radiance measurements at different elevation angles between 2° and the zenith and at several selected azimuth angles free of significant obstacles in the surrounding area. Around 40% of the scattered light measurements were performed at two main azimuthal directions (115° and 315°) (Fig. 2a and 2b). Additional elevation
sequences were performed at azimuth angles 80° relative to the solar azimuth as presented in Fig 2b. The derived tropospheric columns of $NO_2$ are characterized by homogeneous spatial distribution along the effective light paths of the MAX-DOAS (Fig. 2b). Thus, observations for all available azimuthal directions were used for the satellite datasets validation.

The acquired spectral measurements were analysed according to the DOAS method (Platt, 1994; Platt and Stutz, 2008) with the aid of the QDOAS v2.111 software (http://uv-vis.aeronomie.be/software/QDOAS/) developed by the Royal Belgian
Institute for Space Aeronomy (BIRA-IASB) and S[&]T (https://www.stcorp.nl/) (Danckaert et al., 2016). The zenith spectrum of each sequence interpolated at the time of the off-axis measurement was used as the Fraunhofer reference in order to minimize the stratospheric effect in the resulted differential slant column density (dSCD) (Hönninger et al, 2004). The main DOAS analysis settings are summarized in Table 1. $NO_2$ and $O_3$ cross sections have been corrected for the solar Io-effect (Alliwell et al., 2002). An example of $NO_2$ DOAS fitting for a measurement obtained on 7 April 2015 around 07:50 UTC (15:50 local
time) at an elevation angle of 15° and a solar zenith angle (SZA) of about 51° is presented in Fig. 3. The method used in this study to derive the vertical column density (VCD) of $NO_2$ is similar to the one applied in Drosoglou et al. (2017). For the conversion of dSCD into VCD a look-up table (LUT) of AMF was constructed by simulations performed with the uvspec radiative transfer model (RTM), libRadtran version 1.7 (Mayer and Kylling, 2005) using a pseudo-spherical discrete ordinates radiative transfer method (Buras et al, 2011). The aerosol single scattering albedo was assumed to be 0.9, which is a typical
value for urban areas in China (e.g. Li et al., 2007 and references therein), while for the aerosol asymmetry factor a value of 0.7 was used (e.g. Xia et al., 2007). For the surface albedo a value of 0.1 was assumed to be representative of an urban area (Feister et al., 1995; Webb et al., 2000). Moreover, $NO_2$ was assumed to be distributed uniformly in a well-mixed layer extending from the surface up to 1 km height. The vertical profile of aerosol extinction used for the RTM simulations was





extracted from the CALIPSO climatology database (LIVAS, http://lidar.space.noa.gr:8080/livas). Examples of the derived AMFs for different aerosol optical depth (AOD) values at 440 nm are presented in Fig. 4.

## 2.2 Satellite tropospheric $NO_2$ observations

Within the European Space Agency Tropospheric Emission Monitoring Internet Service, www.temis.nl, tropospheric $NO_2$

columns derived from observations by the GOME-2/ MetOp-A, GOME-2/ MetOp-B and OMI/Aura space-borne instruments have been used in this study for validation purposes. The two EUMETSAT MetOp satellites are flying in sun-synchronous orbits with equator crossing times of approximately 09:30 LT and a repeat cycle of 29 days. They were launched in 2006 and 2012 respectively. The default swath width of the GOME-2 scan is 1920 km, which gives a nadir pixel size of 80 km × 40 km (across-track × long-track) and enables global coverage in about 1.5 days. The current primary GOME-2B is operated in this

mode, whereas the older GOME-2A is operated in a reduced swath with a swath width of 960 km and nadir ground pixel size of 40 km × 40 km since June 2013. Further description of the GOME-2 instruments may be found in Munro et al. (2015) and Hassinen et al. (2016). The NASA Aura satellite was launched in 2004 also in a polar orbit and with equator crossing time of 13:30 LT. The Ozone Monitoring Instrument (OMI) is a compact nadir viewing, wide swath (daily global coverage), ultraviolet-visible (270 nm to 500 nm) imaging spectrometer with a foot pixel size at nadir is 13 km ×25 km and, in contrast

to the GOME-2 instruments, this foot pixel size is not constant but increases for the off-nadir positions. Further description of the OMI instrument may be found in Levelt et al. (2006; 2017).

Tropospheric $NO_2$ overpass data from OMI, GOME-2A and GOME-2B satellite sensors have been collected from the www.temis.nl project for the operational period of the MAX-DOAS system for the city of Guangzhou. The tropospheric $NO_2$ columns are derived from satellite observations based on slant column $NO_2$ retrievals performed with the DOAS technique,

and the KNMI combined modelling/retrieval/assimilation approach. The slant columns from the GOME-2 observations are derived by BIRA-IASB whereas the slant columns from OMI by KNMI/NASA. For the retrieval of OMI $NO_2$ product the DOMINO v2.0 algorithm was used (Boersma et al., 2011). The algorithm used for the generation of GOME-2A and GOME-2B products (TM4NO2A version 2.3) is described by Boersma et al. (2004).

Apart from the overpass datasets, monthly mean values averaged on different spatial grids, are also provided within the

www.temis.nl service. For visualisation purposes, such monthly mean gridded data for July 2015 were downloaded, plotted only for the area surrounding Guangzhou and are shown in Fig. 5. The values given are the result of averaging and gridding mostly-clear retrievals (cloud radiance fraction <50%, i.e. cloud fractions approximately <20%). White areas in the plots indicate that no meaningful measurement has been available during the month, because a location was persistently covered by clouds, or because of instrument failure. The gridding procedure accounts for the fraction of a satellite pixel overlapping with

a particular grid cell and so the contribution of every pixel to the monthly mean is weighted with the overlap fraction. Note that the mean tropospheric $NO_2$ column for different grid cells may have very different overlap statistics, i.e. grid cell x may have been covered by only 1 meaningful retrieval, whereas grid cell y may be the average of 30 successful cloud-free retrievals.



## 3 Validation of satellite tropospheric NO₂ datasets

### 3.1 Comparisons of ground-based and space-borne tropospheric NO₂

Observations of tropospheric $NO_2$ from three satellite sensors (OMI, GOME-2A and GOME-2B) have been compared with the tropospheric columns derived by the MAX-DOAS system. For the comparison, we used space-borne retrievals

corresponding to satellite pixel center located within a distance (d) of up to 50 km from the ground-based site and for SZA $\leq$ 75°. In the case of OMI, the closest pixel was selected for the comparisons, whereas in the case of GOME-2 sensors, the average measurement of all pixels within 50 km was calculated. For the OMI dataset, only the pixels unaffected by the so-called "row anomaly" (OMI, 2012) were used and only those corresponding to a cross-track dimension smaller than 60 km. In addition, satellite data were screened for clouds and only observations characterized by cloud fraction (CF) $\leq$ 30% were used.

For the tropical conditions prevailing in Guangzhou this CF value is the minimum acceptable to be used as a threshold for our datasets leading to a sufficient number of data available for reliable comparisons, as smaller CFs are rather rare. Each satellite observation is compared with the mean value of the MAX-DOAS measurements recorded within 1 hour centered at the satellite overpass time. In the next section the effect of the criteria selection in the comparisons of the ground-based and satellite data pairs is discussed at length. The coincidence criteria applied in this section and described above are used as the reference case

in the sensitivity study of section 3.2 (Table 2).

The tropospheric $NO_2$ VCDs retrieved from the ground-based radiance spectra measured at 15° and 30° elevation viewing angles and at all available azimuth viewing angles were used in the comparison with corresponding space-borne observations. The system had been proven to be able to retrieve $NO_2$ with a residual of the order of $10^{-3}$, typical residual values of mini MAX-DOAS systems (Drosoglou et al., 2017). The value of $1 \times 10^{-2}$ has been used as a threshold to filter out disturbed

retrievals under variable conditions, as for example, when fast moving clouds of mist emerge from the nearby river in the Guangzhou area.

Tropospheric $NO_2$ in Guangzhou exhibits large variability both in single measurements and in hourly averages with maximum values exceeding $60 \times 10^{15}$ molecules/cm² (see right plot of Fig. 2). The hourly-averaged values range between 10 and $40 \times 10^{15}$ molecules/cm². Several studies have shown similar tropospheric $NO_2$ VCD levels over other Chinese cities. For example, Jin

et al. (2016) reported monthly-averaged tropospheric $NO_2$ VCDs within the same range over Gucheng in North China for the spring and summer time period. Ma et al. (2013) showed that the daytime mean tropospheric $NO_2$ VCD over Beijing varies from 5 to $133 \times 10^{15}$ molecules/cm² with an average of $36 \times 10^{15}$ molecules/cm² during summertime. The average diurnal variation of the tropospheric $NO_2$ column derived from the MAX-DOAS measurements at the elevation angles of 15° and 30° in Guangzhou is shown in Fig. 6 as hourly averages ($\pm 1\sigma$) over three different MAX-DOAS data subsets, each including the

overpass days of one of the three satellite sensors, i.e. GOME-2A, GOME-2B and OMI. More specifically, the three subsets have been extracted from the whole operational period of the MAX-DOAS instrument considering only the days for which the satellite $NO_2$ overpass data corresponded to the selection criteria mentioned above. A double peak appears at around 10:00 am and 06:00 pm local time, indicating higher anthropogenic emissions. The minimum $NO_2$ levels around local noon reflect





the destruction of $NO_2$ due to photochemical processes (Seinfeld and Pandis, 1998). Unfortunately, our MAX-DOAS dataset covers only spring and summer months and it cannot reveal possibly different diurnal patterns during late-autumn and winter seasons, as observed over industrial areas at mid-latitudes due to different emission strength and $NO_2$ lifetimes (Richter et al., 2005). A double-peak diurnal cycle has been also reported for other Chinese cities in previous studies, e.g. for Beijing (Ma et al., 2013) and Wuxi (Wang et al., 2017a) in spring and summer. A similar pattern for $NO_2$ surface concentration in Guangzhou city has been found by Qin et al. (2009) using measurements performed by a long-path DOAS instrument from 10 to 24 July 2006. The large day-to-day variability mentioned already is evident also in this figure from the calculated large standard deviations (up to $\pm$ 19×10$^{15}$ molecules/cm$^2$). Most of the satellite retrievals seem to fall well within the standard deviations of the MAX-DOAS measurements close in time with the satellite overpass, indicating a generally good agreement in the $NO_2$ levels observed in the Guangzhou area both from space and from the ground. Interestingly, during the overpass days of GOME-2B the tropospheric $NO_2$ levels, both in terms of the average value and the standard deviation, are lower relative to GOME-2A and OMI overpass days. Moreover, only very few of the GOME-2A and GOME-2B overpass data refer to common days of collocations between the two satellite instruments. Possibly, the $NO_2$ spatial distribution over the Guangzhou area during the GOME-2B overpass days is quite smooth and without significant horizontal gradients. This could explain the very good agreement of the GOME-2B averaged tropospheric $NO_2$ column with the MAX-DOAS hourly data in contrast to the slightly lower mean overpass values of the other two satellite instruments, considering also the larger pixel size of GOME-2B (80 km × 40 km) compared to GOME-2A (40 km x 40 km in reduced swath) and OMI (13 km × 25 km at nadir).

The comparison results of the space-borne and ground-based collocations are summarized in Table 2 and presented as time series in Fig. 7 and scatter plots in Fig. 8. These figures as well as the first data column in Table 2 refer to the reference coincidence criteria as described in the beginning of this section. Evidently the number of coincident data pairs is rather small and varies for the three satellite sensors (about double the number for GOME-2B), due to gaps in MAX-DOAS data in conjunction with the different overpass times of the satellites. Also connected to the overpass time are the larger $NO_2$ values reported by MAX-DOAS in the case of the GOME-2 sensors (overpass around 10:00 LT), compared to OMI (overpass around 13:30 LT), as it is evidenced also from Fig. 6. MAX-DOAS and satellite observations are, qualitatively, in good agreement with the calculated correlation coefficients ranging between 0.99 for OMI and 0.76 for GOME-2B. GOME-2B shows a closer to unit slope (0.9) than GOME-2A and OMI (0.75 and 0.78, respectively), despite its large footprint. In addition, GOME-2B shows the smallest mean difference compared to the ground-based measurements, i.e. -1.8×10$^{15}$ molecules/cm$^2$ (-5.7%), probably due to the relatively lower tropospheric $NO_2$ loading observed in the city of Guangzhou by the MAX-DOAS during the overpass days of GOME-2B (Fig. 6). However, we should stress again that these statistics have been derived from a very small number of data points.

In contrast to GOME-2B, our comparison results indicate a systematic underestimation of OMI at higher tropospheric $NO_2$ VCDs (mean bias of -3.52×10$^{15}$ molecules/cm$^2$ or -25.1%) and a similar negative bias of GOME-2A from the ground-based observations (-3.9×10$^{15}$ molecules/cm$^2$ or -10.3%). Our findings are within well agreement with the results of other studies over Chinese areas, which in most cases report underestimation of satellite data. For example, Ma et al. (2013) showed an





underestimation in tropospheric NO$_2$ over Beijing by OMI DOMINO NO$_2$ product between 26 and 38%, depending on the DOMINO algorithm version and the time period, and monthly mean MAX-DOAS NO$_2$ 1.1-1.5 times higher than the DOMINO v2.0 product. They also estimated similar correlation coefficient ranging between 0.91 and 0.93. In the study of Wu et al. (2013) tropospheric NO$_2$ VCDs from mobile DOAS are compared with corresponding OMI retrievals revealing an

underestimation of high NO$_2$ values by the satellite sensor and r of about 0.97. Chan et al. (2015) reported MAX-DOAS NO$_2$ VCDs 2-3 times higher than OMI data over Shanghai during the Shanghai World Expo 2010 and correlation coefficients between 0.67 and 0.93 at four different sites, depending on the air pollution levels. In Wang et al. (2017b), although good consistency is found between the MAX-DOAS and OMI DOMINO v2.0 NO$_2$ retrievals, with r=0.85 and a systematic bias of 1%, for both GOME-2A and GOME-2B a significant overestimation of ~30% is reported and r of 0.57 and 0.45 have been

estimated, respectively.

In general, satellite retrievals represent a weighted average over all the atmospheric layers contributing to the signal observed by the satellite sensor and, thus, suffer from relatively low sensitivity near the surface. This fact, in combination with an unrealistic a priori profile assumption, can lead to an underestimation of high NO$_2$ loadings due to local emission sources in polluted areas, such as the Guangzhou city (Eskes and Boersma, 2003). Also, part of the satellite underestimation can be

attributed to the so-called gradient smoothing effect (Ma et al., 2013) and aerosol shielding effect (Jin et al., 2016, and references therein), as well as to measurements contaminated by clouds.

## 3.2 Effects of the coincidence criteria selection on the comparisons

In order to investigate the effect of the coincidence criteria, the comparisons between the MAX-DOAS and the satellite datasets were repeated for various CF thresholds, namely 0.5, 0.4, 0.2, different time windows for the ground-based data averaging

around the overpass time, i.e. 2, 3 and 4 hours, and different radius limits for the area around the MAX-DOAS station within which the satellite pixel center is located, i.e. 20 km, 30 km and 40 km. In each one case, all the other criteria were kept in their reference value. The statistical results of each of the above comparison cases, including the reference case, are reported in Table 2.

In Fig. 9 bar plots of the statistical results for the different cloud screening thresholds are presented. The agreement between

ground-based and both GOME-2 sensors seems to be only slightly affected by the cloud screening applied (see also Table 2, data columns 1-4), likely due to their large pixel sizes. The average difference of GOME-2B from MAX-DOAS tropospheric NO$_2$ VCD is reduced to -1.80×10$^{15}$ molecules/cm$^2$ (-5.73%) and -0.37×10$^{15}$ molecules/cm$^2$ (0.71%) for CF≤0.3 and CF≤0.2, respectively. In case of GOME-2A, only the intercept shows significant improvement when a CF≤0.2 is used. Interestingly, the corresponding intercept value (CF≤0.2) in case of GOME-2B (1.62×10$^{15}$ molecules/cm$^2$) is much higher compared to the

one for CF≤0.3 (0.40×10$^{15}$ molecules/cm$^2$). However, the intercept cannot be reliably estimated when only a few data pairs (<10 in this case) are available and their dispersion should not be ignored.

In contrast, the choice of CF has a more significant effect on the comparisons of MAX-DOAS data with OMI observations: the available data points are reduced to half and about one third for CF≤0.3 and CF≤0.2, respectively, while metrics are quite



improved. This can be attributed to the higher spatial resolution of OMI compared to GOME-2 instruments, which can be 13 km × 24 km when pointing at nadir. The correlation coefficient and the slope of the linear regression are both improved, respectively, from 0.57 and 0.35 for CF≤0.5, to 0.99 and 0.78 for CF≤0.3, and to 0.99 and 0.79 for CF≤0.2. Moreover, the intercept is improved from $3.03\times10^{15}$ molecules/cm$^2$ (CF≤0.5) to -0.38 and -0.29$\times10^{15}$ molecules/cm$^2$ (CF≤0.3 and CF≤0.2,

respectively), while the mean bias is also reduced to more than half when either the CF≤0.2 or the CF≤0.3 is chosen. These results reconfirm that clouds is an important factor affecting both the satellite and ground-based measurements, and that under clear skies at least the OMI sensor is probing more accurately the tropospheric column of NO$_2$ even at strongly polluted environments like the area around the city of Guangzhou. In the study of Wang et al. (2017b), it is shown that the effects of cloud contamination become significant for CF>40% and >30% for OMI and GOME-2 NO$_2$ product, respectively. Also, Jin

et al. (2016) found significant improvement in the correlation between daily MAX-DOAS and OMI products at Gucheng, a rural site in North China, when more strict cloud screening criteria were applied. More specifically, the correlation coefficient for KNMI OMI DOMINO algorithm increased from 0.74 to 0.90 and from 0.75 to 0.95 for the NASA OMNO2 level 2 product. Depending on the results of our analysis, a relatively low CF threshold (30% or lower) is recommended to be used in future validation studies, especially for OMI products.

The MAX-DOAS data are averaged over a period of time around the satellite overpass time, in order to account for the horizontal gradients of tropospheric NO$_2$ that are smoothed out by space-borne measurements due to the large satellite footprint. The time window selection depends on the satellite ground pixel size and the lifetime of the trace gas under investigation in combination with the prevailing local weather conditions. For simplicity purposes, in this study, fixed values are used for every satellite and the whole collocation datasets. Four different time windows centered at the overpass time have

been investigated, with the reference value included: 1, 2, 3 and 4 hours. The results from the comparison between satellite and MAX-DOAS data are presented in data columns 1 and 8-10 of Table 2. For GOME-2A and GOME-2B the mean difference from the ground-based retrievals is in general lower when a window larger than 1 hour around the overpass time is used and is reduced by more than half for a window of 4 hours. This is in agreement with the large pixel sizes of these two satellite sensors. However, lower correlation between the MAX-DOAS and satellite datasets is derived for larger windows, which

indicates greater dispersion of the data pairs. The effect on the comparisons with OMI is statistically more significant, which is expected due to its smaller pixel size. The correlation coefficient is reduced from 0.99 to values <0.7 and the absolute mean difference is almost three times higher for the time windows of 3 and 4 hours compared to the reference case. Thus, we suggest that a short time window is used in such studies over areas with strong local NOx emissions sources, depending on the satellite pixel size; about 1 hour window in case of OMI and GOME-2A and 1 or 2 hours for GOME-2B.

For the satellite validation two options are possible concerning the selection of the satellite overpass data available; either a temporal average value is calculated from space-borne observations or the closest in distance pixel is selected within a predefined radius from the MAX-DOAS station. The KNMI/NASA OMI overpass dataset used in this study has been already filtered by the distance from the Guangzhou city, i.e. only the closest pixel is reported. From the BIRA GOME-2 datasets an average value of all the pixels within an optimum distance have been used in this study, in order to account for the GOME-2



large pixel size and the random noise of the satellite data. The optimum distance criterion may vary for different satellite sensors and different measurement locations, because it depends on many factors such as the satellite footprint, the trace gas under investigation and its horizontal gradients and the time period selected for the MAX-DOAS data averaging. In the present study, four different radii around the MAX-DOAS location have been investigated, namely 20 km, 30 km, 40 km and the

reference value of 50 km. The statistics estimated for the investigation of the distance criterion selection are reported in data columns 1 and 5-7 of Table 2. The effect of the distance criterion on the comparison of MAX-DOAS retrievals with OMI observations is rather weak. The calculated values of all statistics remain the same for distances 30-50 km. The correlation coefficient, slope and mean bias of OMI from MAX-DOAS are slightly affected for d≤20 km, changing from 0.99 to 0.94, from 0.78 to 0.76 and 3.52 to $3.11\times10^{15}$ molecules/cm$^2$, respectively. Only the intercept value shows a significant improvement

to $-0.01\times10^{15}$ molecules/cm$^2$ (d≤20 km) from $-0.38\times10^{15}$ molecules/cm$^2$. These results are attributed to the fact that the majority of the satellite pixels included in the comparisons are centered within 20 km from the ground-based location. The effect of the radius selection on the GOME-2A sensor is different compared to that on GOME-2B. The comparison of GOME-2A with the MAX-DOAS observations is only slightly affected for a distance limit of 30 km and somewhat improved for d≤20 km. On the contrary, the comparison with GOME-2B seems to be more sensitive to the distance criterion applied on the data pairs'

selection. The number of collocations is one-fourth and two-thirds less for distance ≤30 km and ≤20 km, respectively, compared to the reference case. In addition, the r value is notably improved from 0.76 to 0.93 for d≤20 km, while the best slopes appear for d≤40 km and ≤30 km, i.e. 1.00 and 1.04, respectively, and the absolute mean bias decrease about 6 times for d≤30 km. In general, the distance of the satellite pixel center from the ground-based location depends on the pixel size; for smaller satellite footprints, e.g. OMI and GOME-2A, the pixel center is mostly located within a radius of 20 km, while for

coarser satellite spatial resolution, e.g. GOME-2B, the pixel center can be within a distance of up to 40 km from the MAX-DOAS location. Thus, an upper distance threshold of 30 km seems to be an optimal selection, considering also the statistic results of this study.

## 4 Conclusions

In this study, tropospheric NO$_2$ VCD measurements performed with the MAX-DOAS system of AUTH in Guangzhou, China

are presented and used for the validation of relevant satellite products. The data were collected during a one-year campaign that was held in the framework of the EU FP7 Monitoring and Assessment of Regional air quality in China using space Observations, Project Of Longterm sino-european co-Operation, MarcoPolo project. The MAX-DOAS data are compared with corresponding OMI/Aura, GOME-2/MetOp-A and GOME-2/MetOp-B overpass data, revealing good correlation coefficients, i.e. 0.99, 0.81, 0.76, respectively, and slope values ranging between 0.75 and 0.9. However, the NO$_2$ levels in the troposphere

are underestimated by the satellite sensors on average by 3.5 (25.1%), 3.9 (10.3%) and 1.8 (5.7%) $\times10^{15}$ molecules/cm$^2$, respectively for OMI, GOME-2A and GOME-2B. Similar results have been reported by several studies for OMI observations over other Chinese cities (Ma et al., 2013; Wu et al., 2013; Wang et al., 2017b). However, the agreement of our MAX-DOAS




measurements with GOME-2A and GOME-2B retrievals is better compared to other studies (e.g. Wang et al., 2017b). The underestimation of tropospheric $NO_2$ by satellite sensors can be mainly explained by the relatively low sensitivity of space-borne measurements near the surface, the a priori profile assumed for the AMF calculations, the gradient smoothing effect and the aerosol shielding effect.

Interestingly, GOME-2B shows the smallest underestimation despite its large pixel size (80 km × 40 km). By investigating the diurnal cycles of the ground-based tropospheric $NO_2$ VCD in Guangzhou as an average of the collocation days for each satellite separately, we conclude that the better agreement between the MAX-DOAS and GOME-2B retrievals can be partly attributed to the significantly lower tropospheric $NO_2$ loadings observed during the GOME-2B overpass days. We revealed a diurnal pattern of tropospheric $NO_2$ with two maxima located around late-morning (10:00 LT) and late-afternoon (18:00 LT),
indicating higher anthropogenic emissions, and a minimum close to local noon (~14:00 LT), reflecting photochemical sinks of tropospheric $NO_2$. Similar diurnal variation for the $NO_2$ surface concentration in Guangzhou city has been found by Qin et al. (2009). A double-peak diurnal cycle has been also shown for other Chinese cities, e.g. for Beijing (Ma et al., 2013) and Wuxi (Wang et al., 2017a) in spring and summer.

In order to investigate the effect of the coincidence criteria, the comparisons between ground-based and space-borne
tropospheric $NO_2$ retrievals were repeated for various CF thresholds, different time windows for the averaging of the MAX-DOAS data around the overpass time and different upper limits for the distance of the satellite pixel center from the MAX-DOAS site. The effect of the MAX-DOAS averaging time window on the comparisons with OMI is more significant, probably due to its smaller pixel size. Although the agreement between OMI and MAX-DOAS is worse for larger time windows, for GOME-2 sensors the results are slightly improved. This finding can be explained by the smoothing of the horizontal $NO_2$
gradients due to the GOME-2 large pixel size. On the other hand, the distance criterion has no significant effect on OMI and GOME-2A results because most of the overpass data are located within 20 km from the ground-based station. In case of GOME-2B better slope values and mean biases are achieved for d≤40 km and ≤30 km, while the correlation coefficient is better for d≤20 km. The CF threshold seems to have the most profound effect on the comparisons between satellite and MAX-DOAS datasets, especially in the case of OMI, the underestimation of which is substantially suppressed when more stringent cloud screening is applied (CF≤20%), reducing the average difference to -3.44×10$^{15}$ molecules/cm$^2$ (less than half the value
for CF≤50%) and raising the correlation coefficient to 0.99 and the slope to 0.79 from 0.57 and 0.35, respectively, in case of CF≤50%.

It should be mentioned here that in this study the MAX-DOAS tropospheric $NO_2$ time series covers about 1 year in total, covering only observations during spring and summer months. This means that all the findings are only representative of the
spring-summer seasons and no information is available on the $NO_2$ patterns in the area during late-autumn and winter seasons which are characterized by different emissions strength and $NO_2$ lifetimes.



## Acknowledgements

This study has been conducted within the EU FP7 MarcoPolo project, www.marcopolo.eu. We acknowledge the free use of tropospheric NO₂ column data from the GOME2/MetopA, GOME2/MetopB and OMI/Aura sensors from http://www.temis.nl/airpollution/no2.html. We further acknowledge that the GOME-2 NO₂ data products were generated using

level 1 data developed by EUMETSAT. The overpass tropospheric NO₂ time series for the city of Guangzhou, China, have been disseminated by the EU FP7 MarcoPolo project. The LIVAS products have been collected from the LIVAS database (http://lidar.space.noa.gr:8080/livas), and were produced by the LIVAS team under the European Space Agency (ESA) study contract No. 4000104106/11/NL/FF/fk. The authors would like to thank Prof. Xinming Wang for hosting and supporting Phaethon's operation in the infrastructure of Guangzhou Institute of Geochemistry, Chinese Academy of Sciences. The authors

also would like to thank Zhonghui Huang for his invaluable and continuous support in operating the instrument and overcoming the technical difficulties.

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





**Table 1. Main features of the DOAS analysis.**

| | |
|---|---|
| **Fitting window** | 411-445 nm |
| **Fraunhofer reference spectrum** | Zenith spectrum of each elevation sequence interpolated at time |
| **Polynomial degree** | Order 4 (5 coefficients) |
| **Intensity off-set** | 0th order (constant) |
| **Cross-sections:** | |
| **$NO_2$ (298 K)** | Vandaele et al. (1998) [$I_o$ correction with SCD of $1 \times 10^{17}$ molecules/$cm^2$] |
| **$NO_2$ (220 K)** | Vandaele et al. (1998) [$I_o$ correction with SCD of $1 \times 10^{17}$ molecules/$cm^2$] |
| **$O_3$ (223 K)** | Serdyuchenko et al. (2014) [$I_o$ correction with SCD of $1 \times 10^{20}$ molecules/$cm^2$] |
| **$O_4$ (293 K)** | Thalman and Volkamer (2013) |
| **$H_2O$** | HITEMP (Rothman et al., 2010) |
| **Ring** | Chance and Spurr (1997) |



**Table 2. Statistics of the comparison of tropospheric NO₂ VCD derived from Phaethon and the three satellite sensors, using the reference coincidence criteria (first data column) and for several different cases of CF filtering, time period around overpass and distance limit between the MAX-DOAS station and the satellite pixel center.**

| | Compared to Phaethon NO$_2$ VCD$_{trop}$ | Reference | Cloud Fraction | | | Radius | | | Time Window | | |
|---|---|---|---|---|---|---|---|---|---|---|---|
| | | ≤ 0.3, ≤ 50km, 1 hour | ≤ 0.5 | ≤ 0.4 | ≤ 0.2 | ≤ 40km | ≤ 30km | ≤ 20km | 2 hours | 3 hours | 4 hours |
| **OMI** | Number of data pairs | 11 | 22 | 18 | 8 | 11 | 11 | 9 | 13 | 16 | 19 |
| | Correlation coefficient (r) | 0.99 | 0.57 | 0.62 | 0.99 | 0.99 | 0.99 | 0.94 | 0.94 | 0.62 | 0.66 |
| | Slope | 0.78 | 0.35 | 0.40 | 0.79 | 0.78 | 0.78 | 0.76 | 0.99 | 0.36 | 0.44 |
| | Intercept [x10$^{15}$ molec/cm$^2$] | -0.38 | 3.03 | 2.73 | -0.29 | -0.38 | -0.38 | -0.01 | -3.30 | 5.15 | 3.41 |
| | Mean bias [x10$^{15}$ molec/cm$^2$] | -3.52 | -7.37 | -6.88 | -3.44 | -3.52 | -3.52 | -3.11 | -3.39 | -9.72 | -8.90 |
| | Standard deviation (1σ) [x10$^{15}$ molec/cm$^2$] | 2.11 | 7.11 | 7.05 | 2.45 | 2.11 | 2.11 | 1.53 | 3.27 | 13.37 | 10.91 |
| **GOME-2A** | Number of data pairs | 11 | 13 | 12 | 8 | 11 | 11 | 10 | 11 | 11 | 11 |
| | Correlation coefficient (r) | 0.81 | 0.85 | 0.83 | 0.91 | 0.81 | 0.80 | 0.85 | 0.77 | 0.74 | 0.72 |
| | Slope | 0.75 | 0.79 | 0.77 | 0.83 | 0.75 | 0.73 | 0.80 | 0.74 | 0.76 | 0.77 |
| | Intercept [x10$^{15}$ molec/cm$^2$] | 3.28 | 2.40 | 2.70 | 0.64 | 3.28 | 4.33 | 3.27 | 3.49 | 3.73 | 4.69 |
| | Mean bias [x10$^{15}$ molec/cm$^2$] | -3.91 | -3.51 | -3.72 | -4.14 | -3.91 | -3.56 | -2.23 | -4.19 | -3.05 | -1.48 |
| | Standard deviation (1σ) [x10$^{15}$ molec/cm$^2$] | 9.19 | 8.08 | 8.78 | 7.11 | 9.19 | 9.44 | 8.48 | 9.86 | 10.14 | 10.33 |
| **GOME-2B** | Number of data pairs | 23 | 31 | 28 | 15 | 23 | 17 | 9 | 25 | 25 | 26 |
| | Correlation coefficient (r) | 0.76 | 0.77 | 0.75 | 0.69 | 0.78 | 0.79 | 0.93 | 0.76 | 0.69 | 0.66 |
| | Slope | 0.90 | 0.81 | 0.78 | 0.88 | 1.00 | 1.04 | 1.18 | 0.97 | 0.85 | 0.77 |
| | Intercept [x10$^{15}$ molec/cm$^2$] | 0.40 | 1.53 | 1.73 | 1.62 | -1.38 | -1.27 | 0.31 | 1.55 | 3.72 | 5.10 |
| | Mean bias [x10$^{15}$ molec/cm$^2$] | -1.80 | -2.83 | -2.94 | -0.37 | -1.32 | -0.35 | 4.60 | 0.91 | 0.88 | 0.76 |
| | Standard deviation (1σ) [x10$^{15}$ molec/cm$^2$] | 7.54 | 7.49 | 7.84 | 6.19 | 7.68 | 8.47 | 6.04 | 7.41 | 8.31 | 8.82 |

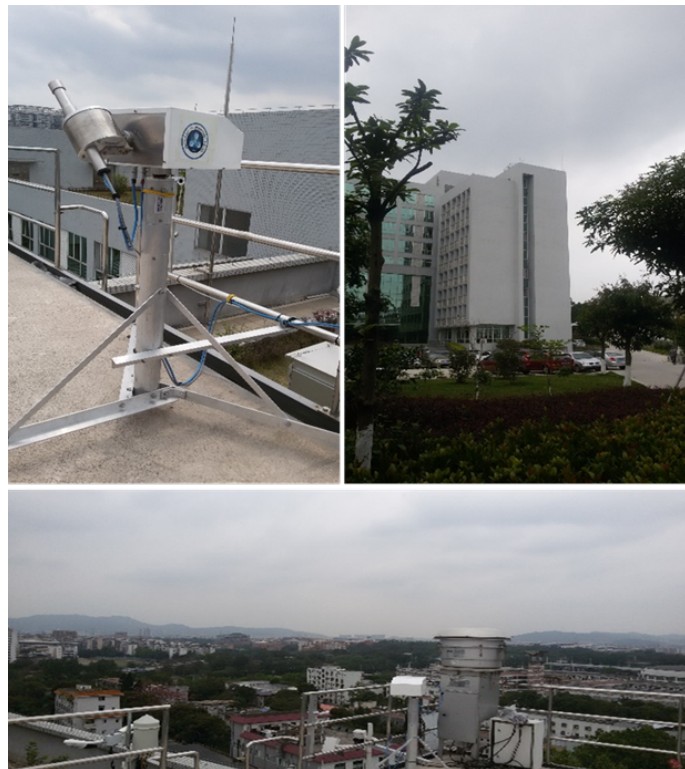

**Figure 1: The tracker and entrance optics of the Phaethon MAX-DOAS system (upper-left) installed on the roof of the Institute of Geochemistry, Chinese Academy of Sciences, in Guangzhou (upper-right) and an overview of the surrounding area towards West (bottom).**





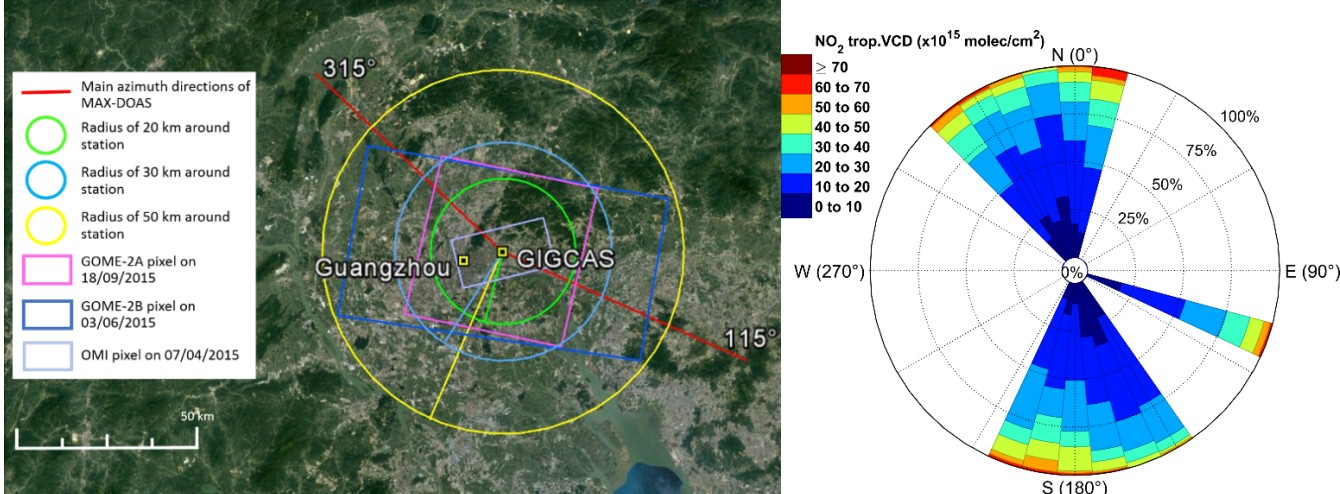

**Figure 2: Left panel: Image of the area around the MAX-DOAS station (GIGCAS) in the Guangzhou megacity, China. The red lines indicate the two main azimuth viewing angles of the MAX-DOAS instrument (115° and 315°). The rectangles outline the GOME-2A, GOME-2B and OMI pixel sizes and positions on 18 September, 3 June and 7 April 2015, respectively. The circles outline areas of different radii around GIGCAS. The image is a courtesy of the Google Earth NASA images. Right panel: Rose diagram showing the frequency of tropospheric NO₂ columns observed at the different azimuth viewing angles as a percentage of the total number of measurements performed in each direction separately.**




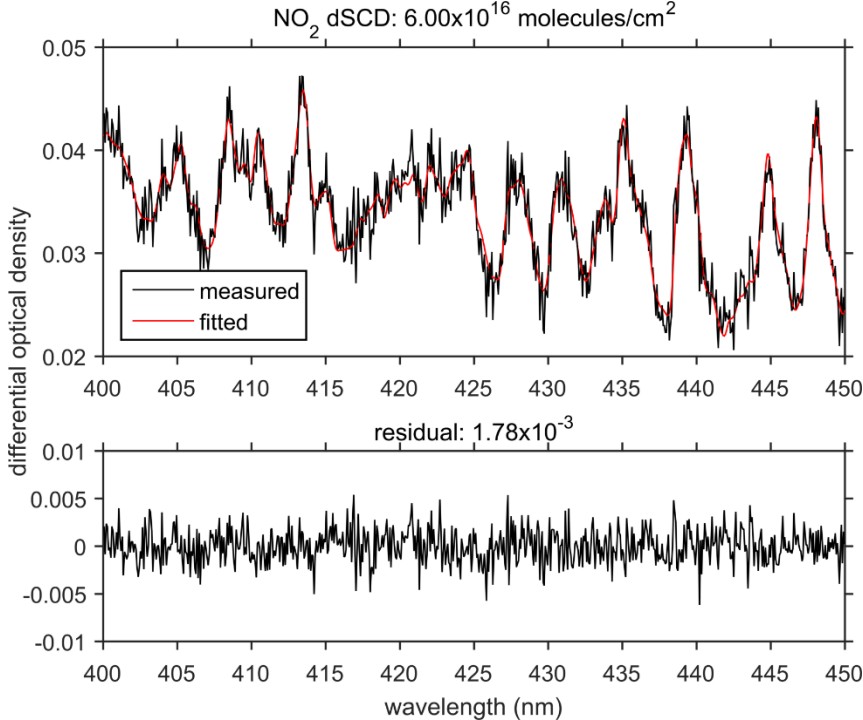

**Figure 3: Example of NO₂ fitting results obtained in Guangzhou on 7 April 2015, around 07:50 UTC (15:50 local time), at an elevation**
5 **angle of 15° and SZA of about 51°. The upper panel shows the measured (black) and the fitted (red) NO₂, and the lower panel shows the residual of the DOAS fit.**




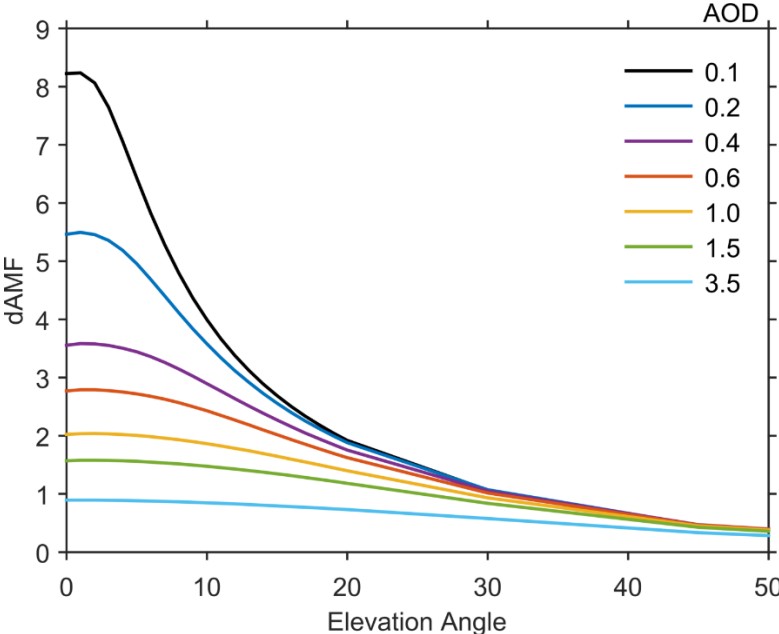

**Figure 4: Examples of calculated tropospheric dAMFs as a function of elevation viewing angle for different values of the aerosol optical depth (AOD) at 440 nm.**



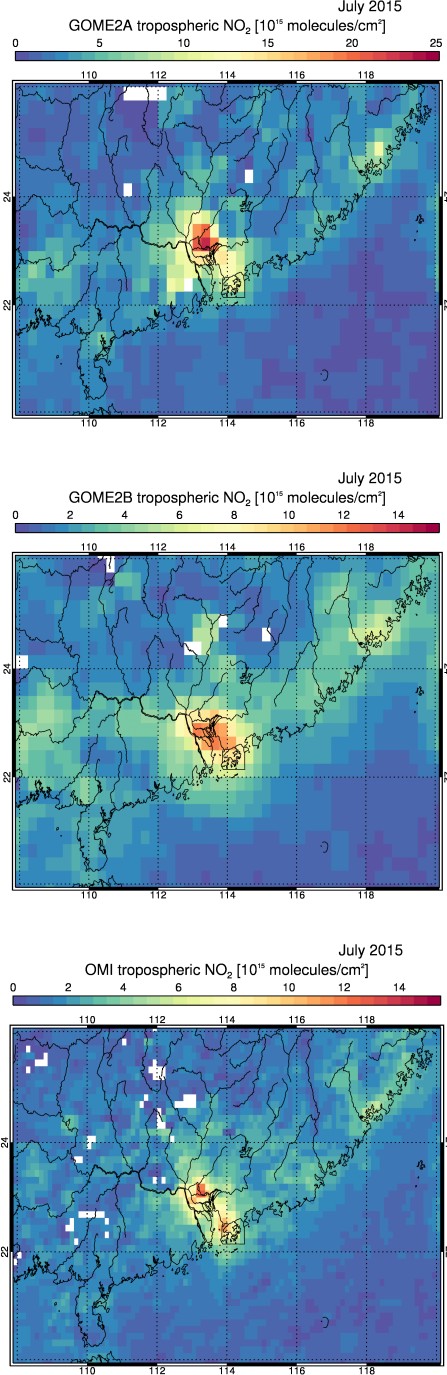

**Figure 5: Monthly averages of GOME-2A, GOME-2B and OMI NO₂ tropospheric VCDs for July 2015 are presented. The GOME-2A monthly tropospheric NO₂ mean values [on a 0.25x0.25° grid] are shown in the upper panel, the GOME-2B, also on a 0.25x0.25° grid, in the middle and OMI/Aura on a finer, 0.125x0.125°, grid in the bottom panel. The NO₂ observations averaged and gridded here correspond to cloud radiance fraction <50%. Note that the color bars have different ranges.**



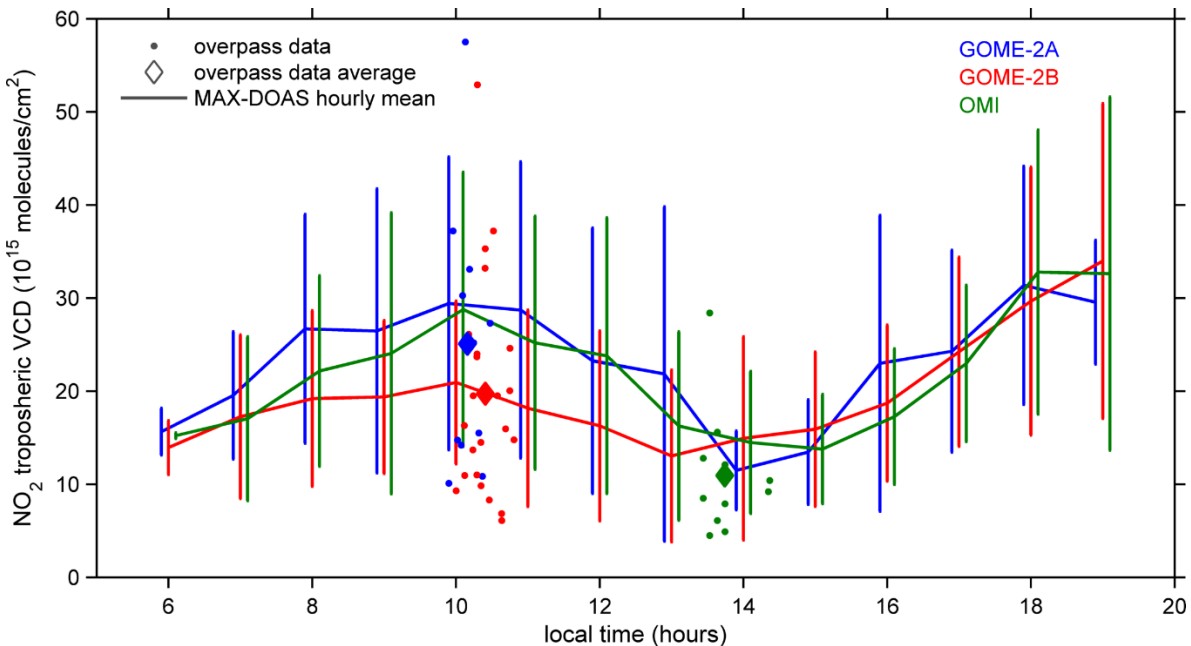

**Figure 6: Average diurnal variation of tropospheric NO₂ columns derived from the MAX-DOAS observations at elevation angles of 15° and 30° in Guangzhou from April 2015 to March 2016. The three lines correspond to the MAX-DOAS hourly mean values resulting only from days with collocated OMI (green), GOME-2A (blue) and GOME-2B (red) data. The error bars indicate the standard deviation of the hourly averages (±1σ). The filled circles represent the NO₂ overpass data of the three satellite sensors that are included in the comparison with the MAX-DOAS measurements and the filed diamonds their average. The symbols are color-coded similarly to diurnals; blue, red and green indicate GOME-2A, GOME-2B and OMI overpass data, respectively.**



**Figure 7: Tropospheric NO₂ in Guangzhou from MAX-DOAS measurements at the elevation angles of 15° and 30° and the satellite sensors OMI (top), GOME-2A (middle) and GOME-2B (bottom) corresponding to cloud fraction ≤0.3. Ground-based measurements are averages of 1 hour centered at the satellites' overpass times. The error bars represent the standard deviation on the mean in all cases except of OMI, where they stand for the measurement error. The grey shaded area stands in for the time period during which the MAX-DOAS was not operating.**



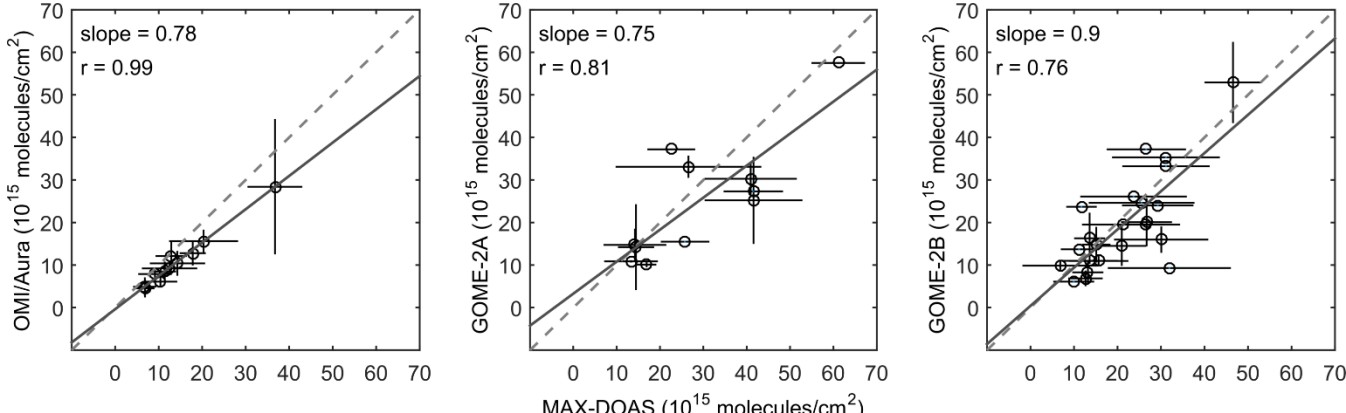

**Figure 8: Scatter plots between the ground-based and satellite tropospheric NO₂ datasets presented in Fig. 7. The error bars represent the standard deviation on the mean in all cases except of OMI, where they stand for the measurement error. The corresponding comparison statistics are presented in the first data column of Table 2.**





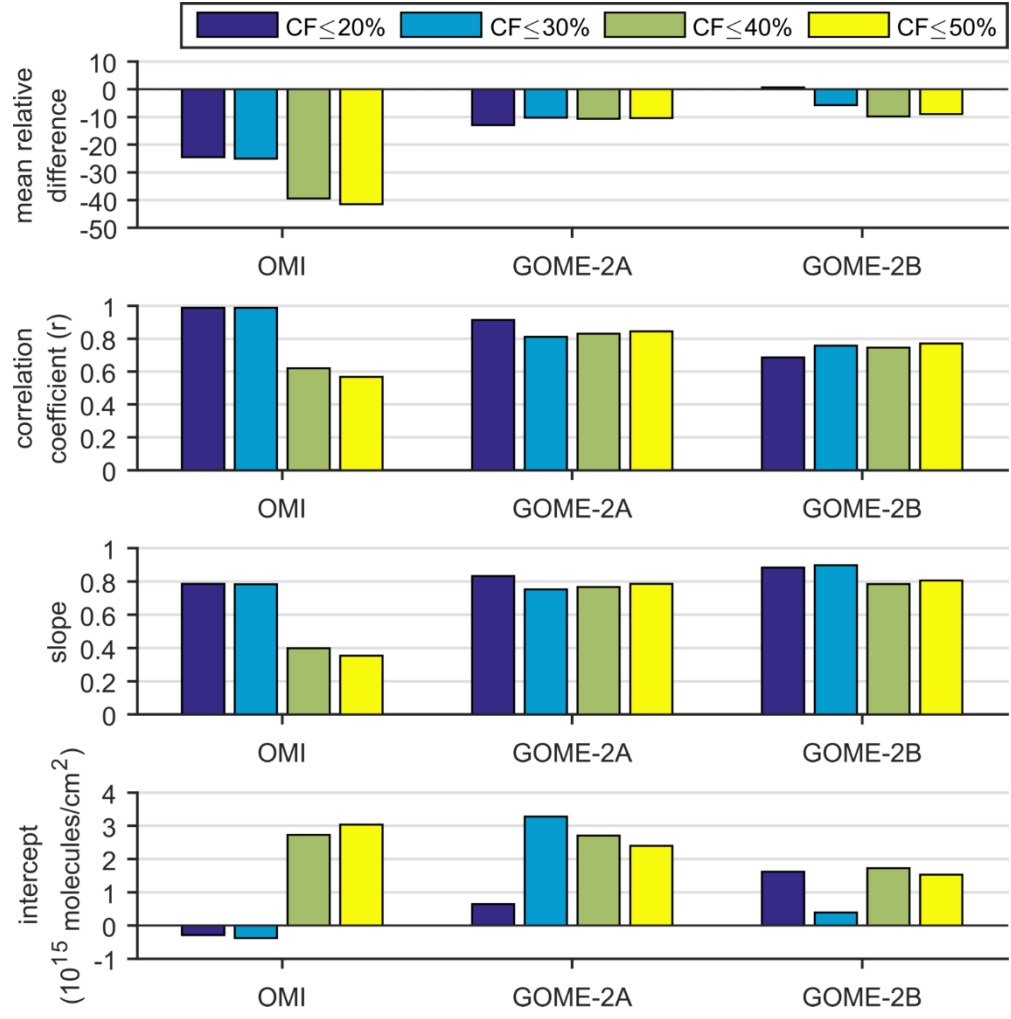

**Figure 9: Bar plots of the statistical results of the comparisons between ground-based and satellite tropospheric NO₂ data pairs for various cloud screening thresholds (four different colors) used as a coincidence criterion.**