# Peer review of "MAX-DOAS NO2 observations over Guangzhou, China; ground-based and satellite comparisons"

_Atmospheric Measurement Techniques, 2017_

## Referee Comment (RC1) · Anonymous Referee #2 · 20 Dec 2017

This paper investigated the tropospheric NO2 vertical column density (VCD) over an urban site in Guangzhou, China using the MAX-DOAS measurements during a campaign for less than a year. The part investing the effect of the main coincidence criteria is interesting and will be useful for the similar analysis in the further. However, I still feel it is a little bit dangerous to make conclusions based on such limited numbers of ground measurements. I suggest the authors to convince readers by either including longer-period data or quantifying uncertainties of the validations.

General comments:

1. Introduction. The authors took too many efforts in describing the importance of NO2, instead of the historical validation using MAX-DOAS. The last paragraph needs to be extended by including more detail introduction of the previous validation works in both China and other regions.

2. Page 7, line 10. The explanation for the better agreement between GOME-2b and ground measurements is not very convincing. "Possibly, the NO2 spatial distribution over the Guangzhou area during the GOME-2B overpass days is quite smooth and without significant horizontal gradients." I suggest providing further evidence (e.g., meteorological parameters) to support this argument, as it is quite an important statement to point out the better agreement of GOME-2b in this paper.

3. Page 7, line 20. As pointed out by the authors themselves, "the number of coincident data pairs is rather small", the reliability of the conclusion is questionable. In addition, the validation result that GOME-2b shows lower bias with ground measurements than the other two sensors is not the same as previous findings, e.g., Wang et al. (2017). Further discussion on uncertainties of this conclusion is necessary.

4. Section 3.2. A summary of the recommended MAX-DOAS settings based on the investigation is helpful for readers.

Specific comments:

1. Page 2, line 27, the bracket is missing in Shao et al., 2009

2. Page 11, line 18. The sentence is too long to read. Please consider rephrasing it.

3. all the x in $NO_x$ should be subscript.

---

## Referee Comment (RC2) · Anonymous Referee #3 · 22 Dec 2017

General comments:

The research paper by Drosoglou et al. presented an investigation of ground-based $NO_2$ measurements by comparing with satellite datasets. The authors use MAX-DOAS $NO_2$ data collected from late March 2015 to mid-March 2016 at Guangzhou, China to compare with three different satellite $NO_2$ datasets, and conclude that all three satellite datasets underestimated tropospheric $NO_2$ concentrations. The authors also investigated effects of some coincident criteria, such as cloud fraction, distance, and averaging time. In general, the paper gives useful and important information about differences in ground-based and satellite $NO_2$ measurements at a specific site (Guangzhou). However, the MAX-DOAS dataset used in this study is too short (and seasonal biased) to draw further solid conclusions (such as local tropospheric $NO_2$ seasonal pattern), or to perform solid satellite validation work.

This paper should be published with major revisions been done.

Specific comments:

Page 4, line11. I can not agree with the author that "the MAX-DOAS observations of tropospheric $NO_2$ were quite sufficient to be used for the satellite data validation". The total coincident measurements were only 22 for OMI, 13 for GOME-2A, and 28 for GOME-2B (found in Table 2). I suggest this paper should not be presented as a satellite validation work, which will need some relative long-term measurements and more work (at least more than one year, without large data gap). I suggest author rephrase relevant parts in the paper, and be concerned with the use of the term -- "satellite validation".

Page 4, line 12. Here it is mentioned the lowest measurement was made with 2°, but why it is not used? On page 6, line 16, it is mentioned that $NO_2$ was retrieved from 15° and 30° measurements.

Page 6, line 18. Please specify which residual is referred to here. Spectral fitting residual, or maybe $NO_2$ dSCDs residual, or something else?

Page 7, line 13-14. Can you find some evidence to support this idea? If we use simple geometry to estimate a coarse horizontal sampling distance of MAX-DOAS instrument (e.g., 15° elevation, 1 km $NO_2$ layer height, as mentioned in your paper), you may find the MAX-DOAS is sampling air mass very close to the site. This makes me feel puzzled, because why GOME-2B (with the largest footprint) could "luckily" measured the days have $NO_2$ smoothly distributed in a larger area. Using the reference criteria group (Table 2), you have 23 coincident measurements with GOME-2B, which doubled the numbers for GOME-2A and OMI. So, maybe this good agreement is not simply due to "luck".

Page 7, line 19. Since the error bars from both MAX-DOAS and satellite data are not small, can you include error weighted fitting? I am wondering if this could improve the results (or show more insightful detials).

Page 8, line 23, Table 2. I found by tightening CF criteria, you have improved r for both OMI and GOME-2A, but (slightly) decreasing trend for GOME-2B. I do not think this is due to over strengthening the criteria, since with CF ≤ 0.2, you still have 15 coincident measurements for GOME-2B, which is about twice the number of coincident measurements from OMI and GOME-2A with CF ≤ 0.2. Do you have any comments on this?

Table 2. Another point is the CF tightening lowered the bias for the OMI and GOME-2B, but (slightly) increased the bias for GOME-2A. Any comments? With small sample size, the changing of bias and/or r should be carefully quantified, before any meaningful conclusion can be made. So, I suggest, for example, maybe calculate the confidence interval for r and bias.

Table 2. The CF criteria part is nicely visualised in Figure 9. Maybe you could perform the same visualisation for the radius and averaging time criteria. Few important information might be overlooked in such large and busy table. For example, why tightening radius criteria led a large jump in GOME-2B mean bias (from -1.8e15 to 4.6e15 molec/cm$^2$)? Any comments?

Page 11, line 5-8. Please provide more evidence (number) to support this idea. Maybe box-and-whisker plots of satellite and ground-based $NO_2$ can show/support that GOME-2B really sampled low $NO_2$ loading condition. It is hard to conclude this by only looking at Figure 6.

General comments:

The sample size in this study might be not good enough for satellite validation; maybe it is to sparse to draw a solid conclusion on the effects of the criteria studied here. However, even if you found the sample size is too small to provide critical results by the end, the process of drawing this conclusion is still valuable. I suggest author spend more time in tightening the conclusions made in the manuscript, and I think this could be a valuable paper.

Technical corrections:

Page 4, line 23. The $I_0$ effect. The subscript should be zero, not "o". Please also correct the ones used in Table 1.

Figure 4. dAMFs is not defined in anywhere. Give unit for the x-axis.

Figure 5. Apparently, Guangzhou is not the only $NO_2$ hotspot on the figures. So, maybe adding location symbol for Guangzhou and a map scale will help the reader to find the city. Please do not use letter x for a multiplication sign.

Figure 9. Give unit (percentage sign) for the top panel y-axis.

Page 12, line 15. Check all your references. Use consistent abbreviations for journals.

Page 15, line 2. A comma is missing after "Atmospheric Research". And please check if the format of page numbers is correct.

Page 16, line 29. This paper has been published on AMT; please do not cite the AMTD version.

---

## Author Comment (AC1) · 9 Mar 2018

**Response to anonymous referee #2**

We would like to acknowledge the referee for their helpful and thorough review. We believe that their comments improved the quality of this work.

Some of the statistics presented in the study have changed as a result of the error weighted linear fitting applied to the collocation data sets.

Our responses (in blue) follow the reviewer's comments (in black italics).

**General comments:**

1. *Introduction. The authors took too many efforts in describing the importance of $NO_2$, instead of the historical validation using MAX-DOAS. The last paragraph needs to be extended by including more detail introduction of the previous validation works in both China and other regions.*

The introduction has been revised accordingly.

2. *Page 7, line 10. The explanation for the better agreement between GOME-2b and ground measurements is not very convincing. "Possibly, the $NO_2$ spatial distribution over the Guangzhou area during the GOME-2B overpass days is quite smooth and without significant horizontal gradients." I suggest providing further evidence (e.g., meteorological parameters) to support this argument, as it is quite an important statement to point out the better agreement of GOME-2b in this paper.*

No evidence could be found to support this statement. The better agreement can be partly attributed to the lower $NO_2$ observed by MAX-DOAS in combination with the larger collocation data set compared to GOME-2A, which improves the metrics of the comparison. The manuscript has been revised.

3. *Page 7, line 20. As pointed out by the authors themselves, "the number of coincident data pairs is rather small", the reliability of the conclusion is questionable. In addition, the validation result that GOME-2b shows lower bias with ground measurements than the other two sensors is not the same as previous findings, e.g., Wang et al. (2017). Further discussion on uncertainties of this conclusion is necessary.*

More statistical analysis results and discussion on them have been included in the manuscript.

4. *Section 3.2. A summary of the recommended MAX-DOAS settings based on the investigation is helpful for readers.*

A summary of the recommended coincidence criteria has been included in section 3.2.

**Specific comments:**

1. *Page 2, line 27, the bracket is missing in Shao et al., 2009*

Brackets have been included.

2. *Page 11, line 18. The sentence is too long to read. Please consider rephrasing it.*

The sentence has been revised accordingly.

*3. all the x in NOx should be subscript.*

The manuscript has been revised accordingly.

---

## Author Comment (AC2) · 9 Mar 2018

**Response to anonymous referee #3**

We would like to acknowledge the referee for their helpful and thorough review. We believe that their comments improved the quality of this work.

Some of the statistics presented in the study have changed as a result of the error weighted linear fitting applied to the collocation data sets.

Our responses (in blue) follow the reviewer's comments (in black italics).

**Specific comments:**

*Page 4, line11. I can not agree with the author that "the MAX-DOAS observations of tropospheric NO$_2$ were quite sufficient to be used for the satellite data validation". The total coincident measurements were only 22 for OMI, 13 for GOME-2A, and 28 for GOME-2B (found in Table 2). I suggest this paper should not be presented as a satellite validation work, which will need some relative long-term measurements and more work (at least more than one year, without large data gap). I suggest author rephrase relevant parts in the paper, and be concerned with the use of the term -- "satellite validation".*

The manuscript has been revised accordingly.

*Page 4, line 12. Here it is mentioned the lowest measurement was made with 2°, but why it is not used? On page 6, line 16, it is mentioned that NO$_2$ was retrieved from 15° and 30° measurements.*

Since the tropospheric columns of NO$_2$ were not derived from profile retrievals, we chose to use the elevation angles of 15° and 30° which are not significantly affected from aerosol. When AMF look-up tables are used for the MAX-DOAS retrievals, uncertainties can be introduced in the VCDs calculated from the observations performed at lower elevation angles, due to the presence of aerosol.

*Page 6, line 18. Please specify which residual is referred to here. Spectral fitting residual, or maybe NO$_2$ dSCDs residual, or something else?*

The spectral fitting residual is mentioned in that sentence. The text has been revised.

*Page 7, line 13-14. Can you find some evidence to support this idea? If we use simple geometry to estimate a coarse horizontal sampling distance of MAX-DOAS instrument (e.g., 15° elevation, 1 km NO$_2$ layer height, as mentioned in your paper), you may find the MAX-DOAS is sampling air mass very close to the site. This makes me feel puzzled, because why GOME-2B (with the largest footprint) could "luckily" measured the days have NO$_2$ smoothly distributed in a larger area. Using the reference criteria group (Table 2), you have 23 coincident measurements with GOME-2B, which doubled the numbers for GOME-2A and OMI. So, maybe this good agreement is not simply due to "luck".*

No evidence could be found to support this statement. The better agreement can be partly attributed to the lower NO$_2$ observed by MAX-DOAS in combination with the larger collocation data set compared to GOME-2A, which improves the metrics of the comparison. The manuscript has been revised.

*Page 7, line 19. Since the error bars from both MAX-DOAS and satellite data are not small, can you include error weighted fitting? I am wondering if this could improve the results (or show more insightful detials).*

Only the error bars of OMI were representing the measurement error. The GOME-2A, GOME-2B and MAX-DOAS error bars were representing the standard deviation of the average value. In the revised manuscript the error bars represent the measurement error in all cases. Thus, the MAX-DOAS errors are quite smaller now. Error weighted fitting has been applied to the collocation data sets and the corresponding figures and statistics have been changed.

*Page 8, line 23, Table 2. I found by tightening CF criteria, you have improved r for both OMI and GOME-2A, but (slightly) decreasing trend for GOME-2B. I do not think this is due to over strengthening the criteria, since with CF ≤ 0.2, you still have 15 coincident measurements for GOME-2B, which is about twice the number of coincident measurements from OMI and GOME- 2A with CF ≤ 0.2. Do you have any comments on this?*

This is probably due to the relatively high variability of the data pairs, which leads to quite wide 95% confidence interval (0.22-0.87) in case of GOME-2B when more stringent CF limit is used. Revisions have been made in the manuscript and the statistical significance of the comparisons is discussed. Also, more statistics are presented in Table 2.

*Table 2. Another point is the CF tightening lowered the bias for the OMI and GOME-2B, but (slightly) increased the bias for GOME-2A. Any comments? With small sample size, the changing of bias and/or r should be carefully quantified, before any meaningful conclusion can be made. So, I suggest, for example, maybe calculate the confidence interval for r and bias.*

95% confidence intervals have been calculated for both r and bias and for each scenario of the coincidence criteria under investigation. All the 95% CI are reported in Table 2. The confidence range for GOME-2A bias is quite wider compared to those calculated for the other two satellite sensors.

*Table 2. The CF criteria part is nicely visualised in Figure 9. Maybe you could perform the same visualisation for the radius and averaging time criteria. Few important information might be overlooked in such large and busy table. For example, why tightening radius criteria led a large jump in GOME-2B mean bias (from -1.8e15 to 4.6e15 molec/cm2)? Any comments?*

This large jump cannot be easily explained. Considering the high r value and the quite larger than unity slope value (1.18), GOME-2B seems to overestimate $NO_2$ columns for high ground-based $NO_2$ observations. However, the 95% confidence range is quite wider compared to those estimated for the OMI and GOME-2A. Bar plots for all collocation criteria have been included in Figure 9.

*Page 11, line 5-8. Please provide more evidence (number) to support this idea. Maybe box-and- whisker plots of satellite and ground-based $NO_2$ can show/support that GOME-2B really sampled low $NO_2$ loading condition. It is hard to conclude this by only looking at Figure 6.*

The text has been revised so that it is clear that lower $NO_2$ levels are observed by MAX-DOAS during the GOME-2B overpass time on GOME-2B collocation days compared to those measured around the same time on GOME-2A and OMI collocation days. Figure 6 can support this statement, because both MAX-DOAS hourly averages and standard deviations around GOME-2B overpass time are lower on GOME-2B days compared to those corresponding to GOME-2A and OMI overpass days. In the following figure box-and-whisker plots are presented. Each satellite box-and-whisker follows the corresponding MAX-DOAS derived. Both GOME-2B and MAX-DOAS observations are lower on GOME-2B collocation days compared to GOME-2A and MAX-DOAS measurements on GOME-2A collocation days. However, this figure is not included in the revised manuscript.

[Figure]

**General comments:**

*The sample size in this study might be not good enough for satellite validation; maybe it is to sparse to draw a solid conclusion on the effects of the criteria studied here. However, even if you found the sample size is too small to provide critical results by the end, the process of drawing this conclusion is still valuable. I suggest author spend more time in tightening the conclusions made in the manuscript, and I think this could be a valuable paper.*

The manuscript has been revised accordingly and more statistics have been included and discussed.

**Technical corrections:**

*Page 4, line 23. The I0 effect. The subscript should be zero, not "o". Please also correct the ones used in Table 1.*

The subscript has been corrected.

*Figure 4. dAMFs is not defined in anywhere. Give unit for the x-axis.*

Revisions have been made.

*Figure 5. Apparently, Guangzhou is not the only NO$_2$ hotspot on the figures. So, maybe adding location symbol for Guangzhou and a map scale will help the reader to find the city. Please do not use letter x for a multiplication sign.*

Changes have been made on the figure and its caption.

*Figure 9. Give unit (percentage sign) for the top panel y-axis.*

The absolute differences are presented now on the figure instead of the % differences.

*Page 12, line 15. Check all your references. Use consistent abbreviations for journals.*

References have been checked.

*Page 15, line 2. A comma is missing after "Atmospheric Research". And please check if the format of page numbers is correct.*

Corrections have been made.

*Page 16, line 29. This paper has been published on AMT; please do not cite the AMTD version.*

The AMTD version has been replaced with the final version.